# The Detrimental Impact of Ultra-Processed Foods on the Human Gut Microbiome and Gut Barrier

**DOI:** 10.3390/nu17050859

**Published:** 2025-02-28

**Authors:** Debora Rondinella, Pauline Celine Raoul, Eleonora Valeriani, Irene Venturini, Marco Cintoni, Andrea Severino, Francesca Sofia Galli, Vincenzina Mora, Maria Cristina Mele, Giovanni Cammarota, Antonio Gasbarrini, Emanuele Rinninella, Gianluca Ianiro

**Affiliations:** 1Department of Translational Medicine and Surgery, Università Cattolica del Sacro Cuore, 00168 Rome, Italy; debora.rondinella@unicatt.it (D.R.);; 2Department of Medical and Surgical Sciences, UOC CEMAD Centro Malattie dell’Apparato Digerente, Medicina Interna e Gastroenterologia, Fondazione Policlinico Universitario Gemelli IRCCS, 00168 Rome, Italy; 3Department of Medical and Surgical Sciences, UOC Gastroenterologia, Fondazione Policlinico Universitario Agostino Gemelli IRCCS, 00168 Rome, Italy; 4Clinical Nutrition Unit, Department of Medical and Surgical Sciences, Fondazione Policlinico Universitario Agostino Gemelli IRCCS, 00168 Rome, Italy; 5Human Nutrition Research Center, Università Cattolica del Sacro Cuore, 00168 Rome, Italy

**Keywords:** ultra-processed foods (UPFs), gut microbiome, gut barrier

## Abstract

Ultra-processed foods (UPFs) have become a widely consumed food category in modern diets. However, their impact on gut health is raising increasing concerns. This review investigates how UPFs impact the gut microbiome and gut barrier, emphasizing gut dysbiosis and increased gut permeability. UPFs, characterized by a high content of synthetic additives and emulsifiers, and low fiber content, are associated with a decrease in microbial diversity, lower levels of beneficial bacteria like *Akkermansia muciniphila* and *Faecalibacterium prausnitzii*, and an increase in pro-inflammatory microorganisms. These alterations in the microbial community contribute to persistent inflammation, which is associated with various chronic disorders including metabolic syndrome, irritable bowel syndrome, type 2 diabetes, and colorectal cancer. In addition, UPFs may alter the gut–brain axis, potentially affecting cognitive function and mental health. Dietary modifications incorporating fiber, fermented foods, and probiotics can help mitigate the effects of UPFs. Furthermore, the public needs stricter regulations for banning UPFs, along with well-defined food labels. Further studies are necessary to elucidate the mechanisms connecting UPFs to gut dysbiosis and systemic illnesses, thereby informing evidence-based dietary guidelines.

## 1. Introduction

The human gut microbiota is a highly intricate community consisting of trillions of microbes residing in our intestinal tract. It co-evolves along with the human host, establishing a symbiosis relationship and plays a fundamental role in key physiological processes, including the maturation and stimulation of immunity, the regulation of nutrient metabolism, protection from pathogens, and even cognitive function [1]. The gut microbiome represents the genomic material of the gut microbiota, although the two terms will be used interchangeably here, due to the equal relevance of microbes and their genomes. The delicate homeostasis of the gut microbiome is increasingly compromised by contemporary dietary habits, with the intake of ultra-processed foods (UPFs) identified as a major contributing factor. UPFs, as defined by the NOVA food classification system, are industrially manufactured products that are typically pre-packaged, energy-dense, and low in nutritional value [2]. UPFs consist of ingredients not commonly found in household kitchens, such as hydrogenated oils, protein isolates, and artificial additives like colorants, emulsifiers, flavor enhancers, and artificial colors. Examples include soft drinks, packaged snack foods, processed meat products, and a variety of ready-to-eat convenience meals. In recent decades, UPFs have become a dominant component of diets in several nations, particularly in industrialized countries [3]. This trend reflects profound shifts in food systems, driven by industrialization and globalization, which have increased the availability and appeal of UPFs due to their affordability, convenience, and long shelf life.

UPFs negatively impact gut health in several ways [3], including reducing the diversity of the gut microbiota, inducing changes in specific bacterial populations, and contributing to the breakdown of the intestinal barrier, which increases intestinal permeability, leading to the condition known as “leaky gut” as shown in Figure 1 [3,4]. A disrupted gut microbiome balance has been linked to various health issues, including obesity, type 2 diabetes (T2DM), cardiovascular disease (CVD), chronic health conditions, and even some mental health disorders [3,5]. Here, we will extensively review the available scientific evidence regarding the impact of UPFs on the human gut microbiome and the intestinal ecosystem. We will first describe ultra-processed foods and the gut microbiome, and then we will explore the underlying mechanisms by which UPFs influence microbial composition and functionality, the documented alterations in gut microbial diversity, and the associated health implications. Additionally, we will also discuss the implications for public health policy and potential strategies for mitigating the detrimental effects of UPFs on gut health.

## 2. Ultra-Processed Foods (UPFs)

### 2.1. The NOVA Classification System

The NOVA classification system is recognized by the Food and Agriculture Organization of the United Nations as a reliable tool for nutrition and public health research and policy development to classify “all foods according to the nature, extent, and purposes of the industrial processes they undergo” [6,7], helping consumers to differentiate foods based on their level of processing and the types of ingredients used. In this system, food is categorized into four groups:(i)NOVA1: Unprocessed or Minimally Processed Foods. These foods undergo minimal processing, such as removing inedible parts and using methods like drying, crushing, and pasteurization, without adding chemicals like sugar, salt, or oils. This category includes cereals such as wheat and rice, legumes, and dairy products like milk and yogurt without added sugars [6].(ii)NOVA2: Processed Culinary Ingredients. This group includes ingredients derived from NOVA1 foods or processed through methods like pressing, refining, and grinding. Examples include salt, sugar, honey, butter, and vegetable oils, which are rarely consumed alone. It also includes items made from two NOVA2 components, such as salted butter and iodized salt. These ingredients are essential for cooking and enhance the flavor and preparation of minimally processed foods [6].(iii)NOVA3: Processed Foods. These foods are created by adding sugar, oil, salt, or other NOVA2 substances to NOVA1 foods, typically featuring two or three ingredients. Preservation and cooking methods, including non-alcoholic fermentation for items like bread and cheese, are used to enhance the durability and sensory qualities of NOVA1 foods. This category includes canned vegetables, canned legumes, salted nuts, cured meats, cheeses, and freshly baked bread [6].(iv)NOVA4: UPFs. These foods are characterized by industrial formulations with five or more ingredients; these products often include additives not commonly found in culinary preparations, alongside sugars, oils, fats, salt, and preservatives. UPFs usually contain little to no NOVA1 components and aim to mimic or mask the sensory qualities of these foods. Examples include non-sugar sweeteners, hydrogenated oils, casein, whey, packaged snacks, sugary drinks, fast food, ready-to-eat meals, processed meats, sweetened breakfast cereals, flavored yogurt, store-bought baked goods, and condiments with additives [6].

Clinicians and epidemiologists are increasingly using the NOVA classification to investigate the potential associations between highly processed food consumption and various health issues such as metabolic disorders, CVD, and certain types of cancers [8,9,10]. Latin American countries have already integrated NOVA into their dietary guidelines [11], and France aims to reduce ultra-processed food consumption by 20% using the NOVA guidelines [12]. This trend suggests NOVA’s application will continue to expand. However, despite some earlier research [13], the system’s robustness and functionality are not well defined, leading to ambiguity and varying interpretations [14], and causing disagreements among experts [15,16]. One source of uncertainty arises from the description of UPFs (NOVA4) as “industrial formulations, typically with five or more and usually many ingredients”. This may have caused some researchers to classify foods with extensive ingredient lists as NOVA4, even if they lacked typical NOVA4 ingredients, such as “substances not commonly used in culinary preparations” [17].

### 2.2. Common Characteristics of Ultra-Processed Foods

UPFs are known for their diverse nutrient content but are often high in added sugars, which increase calorie intake without offering essential nutrients. They also contain unhealthy fats like saturated, trans fats, and hydrogenated oils, which can harm heart health. Additionally, UPFs are typically high in sodium, used as a preservative and flavor enhancer, and contain refined carbohydrates such as white flour. These foods usually lack dietary fiber, crucial for digestive and microbiota health. UPFs contain minimal whole foods and are often heavily processed with artificial additives to enhance taste and sensory appeal [13]. Common additives in UPFs include preservatives, colorings, flavorings, and emulsifiers, which are rarely found in home-cooked meals. Recent studies have highlighted the negative impact of various artificial food additives on gut microbiota and their links to chronic diseases through gut microbial modulation [18].

### 2.3. Processing-Induced Changes

UPFs experience a multitude of industrial procedures to preserve food, as well as enhance taste, texture, and convenience. Understanding these alterations is essential in order to assess the nutritional value of the end product and identify potential health implications. One of the most common processing techniques is the Maillard reaction, which is used to develop sensory attributes like color, flavor, aroma, and texture. While Maillard-reaction products can act as antioxidants, bactericides, antiallergens, and antibrowning agents, they can also function as prooxidants and carcinogens [19]. High-temperature heating during this process can also reduce the nutritional value of proteins. Extrusion is another method, in which a mixture of ingredients is forced through a narrow opening at high temperatures and pressures [20]. This process can induce lipid oxidation, leading to the formation of free radicals and the degradation of unsaturated fatty acids [21]. It is commonly used to create the textures and shapes of snacks, cereals, and processed meats. Hydrogenation involves adding hydrogen molecules to liquid oils to convert them into semi-solid or solid fats [22]. This technique, often used when making margarine, spreads, and baked goods [23], produces trans fats, which are linked to negative health effects such as increased LDL cholesterol levels [24]. Emulsification is a process that uses emulsifiers to create stable mixtures of substances that do not naturally blend, like oil and water [25]. Deep-frying is also commonly used to make fried snacks, fast food, and processed meats. This technique increases the oil content in food, promoting lipid oxidation and potentially generating harmful chemicals like heterocyclic amines [26]. Cryopreservation methods, such as freezing and thawing, are widely used to maintain the quality and extend the shelf life of food, particularly frozen foods and pre-packaged meals [27].

### 2.4. Meal Patterns and Eating Behaviors

UPF consumption (ready-to-eat/heat meals) typically requires less time and effort than unprocessed or minimally processed food intake. Recent research highlights that UPFs could be associated with eating speed, timing, and circadian rhythm disruption. Indeed, a recent study found that late eaters are prone to consume more UPFs and fewer minimally processed foods than early eaters, contributing to increased knowledge of the mechanisms underpinning the association of late eating with adverse cardiometabolic outcomes [28]. A recent meta-analysis also demonstrated that a high intake of UPFs is associated with reduced sleep duration and quality, regardless of age [29]. UPF consumption and sleep showed significant inverse associations. 

## 3. Overview of the Human Gut Microbiome and Gut Barrier

### 3.1. Composition and Diversity of Gut Microbiome

The human gut microbiome is a dynamic ecosystem of microbes that includes bacteria, archaea, viruses, fungi, and protozoa, which reside within the gut. This complex community of microorganisms plays a crucial role in the maintenance of health and the pathogenesis of various diseases [30]. Bacteria are the most ubiquitous and deeply studied component and are classified into five key phyla: Firmicutes, comprising genera such as *Lactobacillus*, *Clostridium*, *Enterococcus*, and *Ruminococcus*; Bacteroidetes, which include genera such as *Bacteroides* and *Prevotella*; Actinobacteria, primarily represented by *Bifidobacteria*; Proteobacteria, which are less prevalent in healthy individuals, however this phylum includes genera such as *Escherichia, Klebsiella*, and *Enterobacter;* and Verrucomicrobia, predominantly represented by *Akkermansia muciniphila* [31]. Under physiological conditions, a homeostatic balance is observed among potentially beneficial bacteria species [31], and Firmicutes and Bacteroidetes constitute the predominant phyla. The ratio between these phyla (F:B) is approximately 1:1, although this may fluctuate depending on various factors, including dietary habits, age, health or disease status, and lifestyle [32]. Conversely, dysbiosis refers to the disruption or loss of this beneficial homeostatic balance [33], although an established definition is still not available. Beyond these dominant phyla, other microorganisms, including methanogenic archaea (*Methanobrevibacter* spp.), various fungi (*Candida* spp., *Saccharomyces* spp.), and viruses (predominantly bacteriophages), contribute to the gut’s functional diversity. These components interact in a complex manner, regulating metabolic pathways, immunological responses, and the gut’s structural integrity [34]. Microbial ecology parameters, such as α-diversity and β-diversity, represent an easy way to assess the balance of the gut microbiome [35]: α-diversity is a taxonomic measure of species diversity within a community that can be expressed by metrics such as richness, which indicate the number of observed species in a community, and evenness, which measures the relative abundance distribution of species within a community. It is widely recognized as a hallmark of a balanced and healthy gut microbiome, as a diverse microbiome is more resilient to external perturbations such as infections, antibiotic use, or dietary changes [36]. For example, β-diversity is a measure of microbial composition between samples (different samples or samples at different timepoints). The intricate interplay between different microbial species in a diverse microbiome also contributes to the production of beneficial metabolites, such as short-chain fatty acids (SCFAs) and the modulation of the host’s immune system. Moreover, a diverse and healthy microbiome may play a crucial role in preventing the overgrowth of opportunistic pathogens, thus maintaining a balanced and healthy gut environment. These functions collectively contribute to maintaining gut homeostasis and promoting overall health [36].

### 3.2. The Functions and Metabolites of the Human Gut Microbiome

The gut microbiota plays a crucial role in various physiological processes, including metabolic functions, immune modulation, and the regulation of gut–brain axis. The gut microbiota is primarily associated with the degradation and fermentation of indigestible dietary components, especially complex carbohydrates, producing key SCFAs such as acetate, propionate, and butyrate, which are essential for gut health and beyond [37]. Firmicutes play a key role in the fermentation of dietary fibers, but Bacteroidetes and Actinobacteria also support gut health through carbohydrate metabolism and complex polysaccharide degradation. *Akkermansia muciniphila*, *Faecalibacterium prausnitzii*, *Clostridium butyricum*, *Prevotella copri*, and *Bacteroides thetaiotaomicron* are among the main SCFA-producing species [38].

#### 3.2.1. SCFAs

SCFAs provide colonocytes with an energy source and maintain the integrity of the intestinal barrier, regulating key processes such as tight junction expression, mucin secretion by goblet cells, the production of antimicrobial compounds by Paneth cells, and also reducing intestinal pH, creating an unfavorable environment for pathogen persistence [33,37].

SCFAs also modulate appetite and improve insulin sensitivity through the secretion of gut hormones such as glucagon-like peptide-1 (GLP-1), peptide YY (PYY), and the binding G protein-coupled receptor (GPR41 and GPR43) [39,40]. Through the same receptors on vascular endothelial cells, SCFAs (in particular, acetate and propionate) influence cardiovascular health, reducing blood pressure [41], lowering cholesterol levels, and modulating hepatic lipid metabolism [42]. Moreover, SCFAs act as crucial systemic mediators that influence metabolism, immune function, and overall homeostasis. SCFAs have profound effects on immune modulation: butyrate promotes anti-inflammatory effects primarily through the suppression of NFkB and the upregulation of PPAR γ by means of epigenetic mechanisms such as the inhibition of histone deacetylases (HDACs) [43] and also the stimulation of the differentiation of Tregs, as well as an increase in levels of anti-inflammatory cytokines like IL-10, the suppressing of pro-inflammatory cytokine production, and the maintenance of epithelial barrier integrity [44].

#### 3.2.2. Immune System Modulation

The gut microbiota has a relevant impact on the shaping of the host immune system, significantly influencing both local and systemic immune responses [45]. The microbial ecosystem interacts with the host through different pathways, including direct interactions with immune cells and the production of metabolites that regulate immune function [46]. One critical route by which the microbiota shapes immunity is through the development and maturation of gut-associated lymphoid tissue (GALT) [47]. Commensal bacteria stimulate the production of antimicrobial peptides, promote the development of isolated lymphoid follicles, and induce the differentiation of T helper cells, particularly Th17 and regulatory T cells (Tregs) [47]. These processes are crucial for maintaining intestinal homeostasis and preventing excessive inflammation. Furthermore, the microbiota influences the development and function of innate lymphoid cells (ILCs), which are essential for mucosal immunity and tissue repair [45]. Recent studies have also highlighted that certain commensal bacteria and fecal microbiota transplantation (FMT) enhance the efficacy of cancer immunotherapy by boosting the function of antigen-presenting cells and enhancing T cell responses [48,49,50]. Conversely, dysbiosis has been associated with various immune-mediated disorders, including inflammatory bowel disease, autoimmune conditions, and allergies [51,52,53,54,55].

#### 3.2.3. Bile Acids (BAs)

The gut microbiota plays a pivotal role in the biotransformation of primary bile acids through mechanisms such as deconjugation, dehydroxylation, and epimerization, leading to the production of secondary bile acids (BAs) [56]. BAs act as signaling molecules by activating nuclear receptors such as the farnesoid X receptor (FXR) and G-protein-coupled bile acid receptor 1 (TGR5) and exhibit many biological activities, including anti-inflammatory properties. For instance, BAs shift neutrophil chemotaxis and macrophage polarization in favor of the anti-inflammatory M2 phenotype, which is required for tissue repair and inflammation resolution and a decrease in pro-inflammatory cytokines such as e IFNγ and IL-6 [57]. In addition, BAs shape the microbial community composition, creating a feedback loop owing to intrinsic bacteriostatic activities [57].

#### 3.2.4. Trimethylamine (TMA) Production

Among the gut microbiota’s numerous metabolic activities, the transformation of compounds containing trimethylamine groups, such as choline, carnitine, and phosphatidylcholine found in animal-derived foods, into trimethylamine (TMA) is of particular interest for human health. TMA is thereafter oxidized in the liver by flavin monooxygenase enzymes (FMO3) to form trimethylamine-N-oxide (TMAO), which has been associated with the pathogenesis of CVD and metabolic syndrome (MetS) [58]. The synthesis of TMA is primarily attributed to Firmicutes and Proteobacteria, two phyla frequently increased in pro-inflammatory conditions [59]. In particular, *Klebsiella pneumoniae*, *Clostridium sporogenes*, *Escherichia fergusonii*, and *Desulfovibrio desulfuricans* are among the principal TMA-producing bacterial species [60].

#### 3.2.5. Gut–Brain Axis

Accumulating evidence indicates that the gut microbiota exerts powerful influences over the central nervous system (CNS) via the gut–brain axis [61]. This bidirectional communication system of hormonal, neural, and immunological pathways connects the gastrointestinal tract with the brain [62]. One of the mechanisms through which microbial species can affect the host is by modulating the synthesis of neurotransmitters including serotonin (5-HT) and gamma-aminobutyric acid (GABA), which play an essential role in mood regulation, cognitive function, and mental well-being. *Lactobacillus* and *Bifidobacterium* species are well known for their association with increased GABA production [63], while *Enterococcus* species are involved in 5-HT synthesis [64].

#### 3.2.6. Microbiota–Gonad Axis

A bidirectional communication between the gut microbiota and the gonads, termed the “microbiota-gonad axis,” has been identified [65]. This interaction involves endocrine, immunological, and metabolic mechanisms that directly influence reproductive function [65]. The gut microbiota strongly impacts the metabolism of sex hormones, such as estrogens and testosterone, via direct enzymatic actions and the indirect modulation of host physiology [66]. This process involves β-glucuronidase activity, essential for the hydrolysis of conjugated estrogen metabolites [67]. The gut microbiota also significantly influences sperm quality parameters in males, including motility, concentration, and morphology. Beneficial metabolites produced by intestinal bacteria, such as SCFAs, have demonstrated a positive impact on sperm health. [65].

#### 3.2.7. Gut Microbiota’s Role in Defense and Metabolic Health

The gut microbiota provides defense against pathogen colonization through various mechanisms, including competitive exclusion, wherein commensal microorganisms outcompete pathogens for nutrients and epithelial binding sites [68]. Additionally, it produces antimicrobial compounds such as bacteriocins and bile acids, while simultaneously supporting intestinal barrier integrity to prevent pathogen translocation into the circulatory system [69]. Furthermore, the microbiota serves a crucial function in synthesizing indispensable vitamins, notably vitamin K and specific B vitamins, including B12, B6, and folate, which are fundamental to numerous metabolic processes within the host organism, encompassing mechanisms such as blood clotting, DNA synthesis, and cellular energy production [70,71]. The mutualistic association between the microbiota and the host organism underscores the significant role of microbial populations in sustaining metabolic equilibrium and overall physiological function [72].

#### 3.2.8. Determinants of Gut Microbiota Composition

The gut microbiota composition shows a considerable interpersonal variability, influenced by a multifaceted interplay of factors encompassing dietary habits, chronological age, genetic predisposition, pharmacological interventions (e.g., antibiotic administration), and environmental influences [73]. While the microbiota typically achieves a comparatively stable state during adulthood, it maintains a dynamic ability to respond to modifications in dietary intake and lifestyle choices. Of particular significance is the impact of ultra-processed food consumption, characterized by elevated levels of refined sugars, detrimental fats, synthetic additives, and insufficient fiber content [4]. Such dietary patterns can perturb the equilibrium of microbial communities, fostering the expansion of potentially harmful taxa while concurrently diminishing the prevalence of beneficial commensal organisms [4]. This microbial imbalance, or dysbiosis, is linked to unfavorable metabolic and inflammatory consequences, underscoring the pivotal role of dietary quality in preserving gut microbial homeostasis and, by extension, overall physiological well-being [74,75].

### 3.3. Gut Barrier

The gut barrier is a complex, multi-layered structure that plays a crucial role for preserving intestinal health and overall well-being. This barrier is structurally composed of the gut microbiota, which competes with pathogens and influences immunity; a mucus layer with antimicrobial properties; epithelial cells that form both a physical and immune barrier; and GALT, which is responsible for monitoring immune activity [76]. Tight junctions between epithelial cells regulate paracellular permeability, ensuring the gut barrier’s integrity. When the gut barrier is compromised, it can lead to a wide range of gastrointestinal and systemic disorders, including type 1 diabetes, allergies, and neurological disorders [76]. Recent studies have elucidated the impact of the gut microbiome on barrier function. Beneficial bacteria contribute to the barrier’s integrity by releasing SCFAs and other metabolites that promote mucus synthesis and tight junction formation [37,77]. Dysbiosis, otherwise, can result in increased intestinal permeability, referred to as “leaky gut”, which has been associated with several clinical conditions such as inflammatory bowel disease, metabolic abnormalities, and neurological disorders [78]. UPFs, stress, and exposure to environmental toxins all have a negative impact on gut barrier function [79,80,81,82]. Understanding the complex connections between these variables and the gut barrier is critical for developing targeted treatment approaches to promote intestinal health and prevent illnesses.

## 4. UPFs-Driven Alterations of Gut Microbiome

An increasing body of evidence shows that UPF consumption can induce significant alterations in the composition and function of the gut microbiome [79,82,83]. UPFs, characterized by a high content of additives, preservatives, emulsifiers, and artificial compounds, are associated with the disruption of the microbial ecosystem, characterized by the proliferation of pro-inflammatory strains and a reduction in microbial α-diversity [83]. At the functional level, the consumption of UPFs has been linked to the decreased production of SCFAs and other protective metabolites by the microbiota [79]. The observed changes in microbial composition and function might have far-reaching impacts on human health, resulting in low-grade, systemic inflammation and oxidative stress, which are crucial elements in the onset of several types of chronic diseases attributed to UPF dietary consumption [84,85]. A Spanish study assessed the gut microbiome of adult subjects who regularly consumed < 3 servings of UPFs daily (96 participants) and of those who consumed >5 servings of UPFs daily (90 participants) [83]. There were no variations in α-diversity measures among women across UPF intake groups, while α-diversity decreased in men who consumed a higher quantity of UPFs, suggesting that UPFs may affect microbiota composition differently in women and men [83]. Individuals who consumed high levels of UPFs exhibited an elevated presence of potentially harmful bacterial groups compared to those with low UPF intake [83]. These bacteria included *Granulicatella*, which is linked to metabolic disorders, as well as *Blautia*, *Carnobacteriaceae, Bacteroidaceae, Peptostreptococcaceae, Bacteroidia*, and *Bacteroidetes* [83]. Conversely, the high UPF consumers showed decreased levels of *Lachnospira* and *Roseburia*, which are known producers of short-chain fatty acids (SCFAs) [83]. Commonly utilized emulsifiers, including carboxymethylcellulose (CMC), polysorbate 80 (P80), carrageenan, and gums, are known to alter gut microbiota composition, thereby promoting a pro-inflammatory microbial environment that may contribute to the development of metabolic disorders such as obesity and T2DM [86,87]. These emulsifiers diminish the prevalence of beneficial bacteria such as *Faecalibacterium prausnitzii* and *Akkermansia muciniphila*, which possess anti-inflammatory properties and adversely affect the intestinal mucus layer, resulting in increased permeability (“leaky gut”) and bacterial translocation into the bloodstream, potentially eliciting systemic inflammation [88,89].

Animal studies further reveal that exposure to emulsifiers leads to a decrease in microbial diversity, promotes the growth of opportunistic pathogens (such as *Escherichia coli*), and shifts gut microbiota metabolism towards a pro-inflammatory state [90,91]. P80 enhances the vulnerability of the small intestine of murine models by promoting dysbiosis, increasing Gammaproteobacteria abundance and decreasing α-diversity in the small intestine [92]. CMC and P80 impact gut microbiota gene expression, leading to the increased production of lipopolysaccharide (LPS) and flagellin, both known as inflammatory molecules that can damage the gut barrier [91]. Prolonged exposure to these emulsifiers also results in a reduction in mucus layer thickness, thereby compromising its protective function against pathogens. [91]. An ex vivo and in vivo study investigated direct P80 and CMC using a mucosal simulator of the human intestinal microbial ecosystem (M-SHIME), and then emulsifier-treated microbiotas were transferred to germ-free mice to assess potential host effects [82]. Both CMC and P80 altered microbiota composition and increased the pro-inflammatory potential; the germ-free mice colonized with emulsifier-treated microbiotas exhibited low-grade inflammation, increased metabolic markers, higher fasting blood glucose levels and increased adiposity, indicative of MetS [82]. A randomized controlled study (RCT) conducted on human participants investigated the effects of CMC on the gut microbiome and metabolic processes [79]. Participants who consumed CMC exhibited changes in gut microbiota composition, including decreases in *Faecalibacterium prausnitzii* and *Ruminococcus* sp., and increases in *Roseburia* sp. and Lachnospiraceae in the fecal metabolome, particularly with reductions in SCFAs and free amino acids [79]. Another study in healthy volunteers showed that the consumption of sweeteners such as sucralose and aspartame at high concentrations can induce apoptosis in intestinal epithelial cells, while their consumption at low concentrations reduces the expression of claudin 3, increasing the permeability of the intestinal barrier [18]. These sweeteners also stimulate the expression of pro-inflammatory mediators such as LPS and the activation of the NF-κB pathway, which is crucial for the progression of intestinal inflammation [18].

Additionally, UPFs tend to be high in saturated fat. This is because, during the industrialization process, ingredients such as refined vegetable oils, butter or animal fats are added to increase their content. These ingredients are added to improve taste, texture, and shelf life, but the high saturated fat content can be detrimental to health, especially if consumed in excess, as it can increase the risk of CVD [93]. Diets high in fat also impact the regulation of gut bacteria populations, causing a decrease in *Bacteroides*, *Verrucomicrobia*, *Eubacterium rectale*, *Clostridium coccoides*, and *Bifidobacterium* groups, while simultaneously increasing the proportion of *Firmicutes* and *Proteobacteria* [84], and increase pro-inflammatory cytokines (IL-1, IL-6, and TNF-α) [94]. In addition, saturated fat increases the concentration of plasma endotoxin derived from Gram-negative bacteria and promotes hyperinsulinemia and excessive lipid accumulation in the liver and adipose tissue [84]. The regular consumption of UPFs negatively affects both the gut microbiota and metabolic profiles, with a significant reduction in advantageous species which play crucial roles in maintaining gut health by fermenting fibers and generating SCFAs [95].

## 5. Impact of UPF-Induced Dysbiosis on Human Health

As previously outlined, a diet rich in UPFs leads to gut microbiome imbalance, impaired intestinal barrier function, and enhanced gut permeability. This allows harmful bacterial products, such as LPS, to enter the bloodstream, triggering a status of systemic low-grade inflammation [31]. This process contributes to the rising incidence of noncommunicable chronic disorders (NCDs), which are complex conditions characterized by persistent, low-grade inflammation and disrupted gut microbiota balance. These disorders include cardiometabolic conditions like MetS, T2DM, and CVD. Chronic gastrointestinal conditions, such as inflammatory bowel disease (IBD) and irritable bowel syndrome (IBS), are also part of this group. Furthermore, neurological and psychiatric conditions are linked to these inflammatory processes [75].

### 5.1. Cardiometabolic Disorders

The metabolic syndrome is a pathophysiological condition of metabolism which fulfils at least three of the five criteria concerning the following aspects: central obesity, hypertension, hyperglycemia, hypertriglyceridemia, and low serum HDL. It is associated with an increased risk of T2DM and CVD [95]. A consolidated body of evidence has shown a link between obesity, metabolic syndrome and gut microbiome imbalance, characterized by reduced microbial diversity and an increased capacity to extract energy and produce harmful compounds, such as TMAO [96,97]. UPF-induced microbiome alterations have been linked to increased cardiometabolic risk: T2DM appears to be influenced by an elevated ratio of Firmicutes to Bacteroidetes, which is associated with a pro-inflammatory state [75]. Moreover, *Akkermansia muciniphila*, which is less abundant in individuals consuming diets rich in UPFs, improves insulin sensitivity, insulin levels, and weight control in obese, insulin-resistant subjects, and is associated with a lean body type in both animal and human studies [98]. The microbiome also produces harmful compounds that may increase cardiovascular risk: the gut microbiome plays a role in metabolizing several aminoacides, producing TMA, which is then converted to TMAO. TMA is primarily produced by Firmicutes and Proteobacteria, two bacterial groups often increased in gut dysbiosis [59]. Higher TMAO production is linked to an increased risk of cardiometabolic diseases [99]. For instance, a prospective study of patients undergoing elective coronary angiography showed that fasting TMAO levels were directly related to major adverse CVD events over a 3-year follow-up period [100]. Additionally, patients with stable coronary artery disease and with higher TMAO levels had an increase in 5-year all-cause mortality, possibly due to TMAO’s atherogenic effects [101].

### 5.2. Inflammatory Bowel Disease (IBD)

Inflammatory bowel disease, mainly Crohn’s disease (CD) and ulcerative colitis (UC), are conditions characterized by chronic and recurrent inflammation of the intestinal mucosa. The onset of IBDs is linked to multiple factors, including genetic predisposition and exposure to detrimental environmental conditions. [102] An imbalance in the gut microbiota of patients with IBD has been demonstrated to contribute to disease progression through various mechanisms, such as compromised function and enhanced permeability of the intestinal barrier, as well as increased immune system activation [51]. Individuals with IBD exhibit both qualitative and quantitative alterations in their gut microbiota compared to healthy subjects: a-diversity is diminished in both CD [103] and UC [104] patients, accompanied by a reduction in anti-inflammatory bacterial species, including *Roseburia hominis*, *Akkermansia muciniphila*, *Faecalibacterium prausnitzii*, and *Eubacterium rectale*. Furthermore, there is an increase in pro-inflammatory bacterial species, such as *Escherichia coli*, *Ruminococcus gnavus,* or *Fusbobacterium* spp. [105]. These alterations have been notably associated with disease exacerbations and active inflammation, while patients with quiescent IBD display a bacterial profile more closely resembling that of healthy individuals [52]. Changes in the bacterial composition of subjects with IBD also alter key metabolic pathways. Notably, in IBD patients, three main pathways are affected: SCFA production [106], bile acid metabolism [53], and tryptophan metabolism [54]. These alterations contribute to the promotion and sustainment of intestinal inflammation [107].

### 5.3. Cancer Risk

As previously noted, both cardiometabolic disorders and IBD are characterized by a chronic inflammatory state induced by dysbiosis, which plays a crucial role in various stages of cancer development [107]. However, considering that the increasing incidence of colorectal cancer (CRC) has generally followed the adoption of the Western lifestyle [86], the Western diet has emerged as a primary factor influencing the global incidence of CRC [108]. An increasing body of evidence shows differences in the fecal microbiome between patients with CRC and healthy individuals [109]. Similarly to IBD, the dysbiosis associated with CRC is characterized by reduced microbiome diversity, the loss of anti-inflammatory bacteria, and an increase in pathobionts. A specific bacterial signature has been identified in several cohorts, demonstrating high specificity for CRC [110]. *Fusobacterium nucleatum*, *Bacteroides fragilis*, and *Parvimonas micra* are among the primary species linked to adenomas and CRC. These taxa are typically present in very low abundance or are absent in the fecal microbiota of healthy individuals, but their proliferation may promote the onset of dysplasia and mucosal invasion, as demonstrated by numerous animal studies [109]. These pro-carcinogenic bacteria may exert their effects by inducing an inflammatory microenvironment or through direct oncogenic action, producing various metabolites such as toxins or reactive oxygen species that are capable of causing DNA damage [108].

### 5.4. Mental Health and Cognitive Function

Cognitive function and mental health are closely linked to environmental factors such as diet, lifestyle, and gut microbiota composition [111]. The “gut-brain axis” describes this bidirectional interaction between the gut and the brain, influencing homeostasis through digestion, microbial metabolites, and overall health [112]. Due to this intricate connection, alterations in gut microbiota composition can lead to psychopathological conditions such as Alzheimer’s disease (AD), Parkinson’s disease (PD), and depressive disorder [75,112,113]. Taken together, these findings suggest that consuming ultra-processed foods could have harmful effects on mental well-being, possibly through inflammatory pathways and disturbances in the gut–brain connection [114]. A 2023 meta-analysis, which examined 26 observational studies investigating the link between UPF intake and mental health disorder risk, indicated that UPF consumption was linked to a higher risk of depression, but not anxiety. Furthermore, a dose–response analysis highlighted a positive linear relationship between UPF intake and depression risk [115]. Moreover, a comprehensive umbrella review conducted in 2024, analyzing 39 meta-analyses to explore the connection between UPFs and various health outcomes, uncovered highly suggestive evidence of a correlation between UPF consumption and both depression and common mental disorders [5]. Studies demonstrate that WD accelerates beta-amyloid accumulation in animals [116] and that diet can both induce and exacerbate AD pathology in humans and rodents [75]. Prebiotics and probiotics appear to improve cognitive conditions in PD patients, emphasizing the role of diet and the gut microbiota in disease pathogenesis [113]. Additionally, WD contributes to depression via nutrient–microglia and gut–immune interactions [112]. Major depressive disorder (MDD) patients exhibit reduced gut microbiota richness and diversity compared to healthy individuals, with altered levels of Bacteroidetes, Firmicutes, Proteobacteria, and Actinobacteria [112].

## 6. Strategies to Mitigate the Negative Effects of UPFs on Human Health

UPFs exert a negative impact on gut health, primarily by altering gut microbiota composition and function. The mitigation of these effects requires a multifaceted approach encompassing dietary modifications, targeted supplementation, and lifestyle changes.

### 6.1. Dietary Strategies

Reducing the consumption of UPFs is a cornerstone strategy for preserving intestinal health. UPFs often contain high levels of refined sugars, unhealthy fats, and synthetic additives, which can disrupt the gut microbiota [117]. Transitioning to a diet centered on whole and minimally processed foods, such as fruits, vegetables, whole grains, lean proteins, and fermented foods, supports microbial diversity and resilience [118].

Particular attention should be given to dietary fiber intake. High-fiber foods, such as legumes, nuts, seeds, and whole grains, provide substrates for beneficial gut bacteria, promoting the production of SCFAs like butyrate, acetate, and propionate. These SCFAs are pivotal for maintaining intestinal barrier integrity and modulating inflammation [37]. Another effective strategy is the incorporation of fermented foods, such as yogurt, kefir, sauerkraut, kimchi, and miso (Table 1). These foods are rich in probiotics—live beneficial bacteria that can positively influence gut microbiota composition. Regular consumption of these foods can counteract the dysbiotic effects of UPFs, restoring microbial balance and improving gut health [119].

### 6.2. Probiotic and Prebiotic Supplementation

Probiotic supplementation has emerged as a targeted intervention for mitigating the adverse effects of UPFs on gut microbiota (Table 1). Specific strains of probiotics, such as *Lactobacillus rhamnosus*, *Bifidobacterium longum*, and *Saccharomyces boulardii*, have demonstrated potential in reducing gut inflammation, enhancing mucosal immunity, and restoring microbial diversity [120]. The choice of probiotic strain should be tailored based on individual needs and the nature of dysbiosis induced by UPFs. Prebiotics, non-digestible food components that stimulate the growth of beneficial bacteria, are another critical intervention. Common prebiotics include inulin, fructooligosaccharides (FOSs), and galactooligosaccharides (GOSs). These compounds selectively feed beneficial bacteria, promoting their proliferation and the production of SCFAs. Combining probiotics with prebiotics (synbiotics) may have synergistic effects in reversing UPF-induced gut microbiota disturbances [121]. *Akkermansia muciniphila* has emerged as a promising next-generation probiotic due to its potential to enhance intestinal barrier function, modulate immune responses, and improve metabolic health [89,122,123]. The daily oral administration of *Akkermansia muciniphila* in a mouse model exposed to CMC and P80 for nine weeks counteracted the emulsifiers-induced inflammatory effects, preventing mucosal immune activation and epithelial damage [89].

### 6.3. Lifestyle Factors (Exercise, Stress Management, and Sleep)

Lifestyle factors play a significant role in gut health and can mitigate the adverse effects of UPFs. Regular physical activity is associated with increased microbial diversity and the promotion of SCFA-producing bacteria. Exercise also enhances intestinal barrier function and reduces systemic inflammation, counteracting some of the deleterious effects of UPFs [124]. Stress management is another crucial element, as chronic stress can disrupt gut microbiota through the gut–brain axis. Techniques such as mindfulness, meditation, and cognitive–behavioral therapy can help regulate stress responses, thereby supporting a healthy gut environment [125]. Sleep quality and duration are also integral to gut health. Disrupted sleep patterns can alter gut microbial composition and promote dysbiosis. Establishing consistent sleep routines and prioritizing sleep hygiene can help maintain microbial balance, complementing dietary and supplementation strategies [126].

### 6.4. Implications for Public Health Policy

The widespread consumption of UPFs poses significant public health challenges. To address their impact on intestinal health, policy interventions should focus on dietary guidelines, food labeling and marketing regulations, and broader food environment interventions (Table 1).

Incorporating the degree of food processing into dietary recommendations is essential for effective public health guidance. Traditional dietary guidelines focus on nutrient composition but often neglect the impact of food processing on gut health. Updated guidelines should advocate for the minimization of UPF consumption and should emphasize the importance of whole and minimally processed foods. Educational campaigns targeting both healthcare professionals and the general public can further enhance awareness of the gut health implications of UPFs. These campaigns should highlight the benefits of dietary fiber, fermented foods, and diverse dietary patterns for microbial health [127].

### 6.5. Food Labeling and Marketing Regulations

Clear and transparent food labeling is a powerful tool for informing consumer choices. Front-of-pack labels (FoPLs), such as traffic light systems or warning symbols, can effectively communicate the degree of processing and the presence of additives in food products. Policymakers should mandate such labeling to enable consumers to make informed decisions [128]. Marketing restrictions for UPFs, particularly those targeting children, are another critical area of intervention. Evidence suggests that the aggressive marketing of UPFs contributes to poor dietary habits and increased consumption. Implementing stricter regulations on advertising, including digital platforms and packaging designs, can help curb the appeal of UPFs [129].

### 6.6. Food Environment Interventions

Transforming food environments to prioritize health-promoting options is essential. School food policies, for instance, can play a pivotal role in shaping dietary habits from an early age. Mandating the provision of minimally processed, fresh foods in schools can help reduce UPF consumption among children and adolescents [130]. Urban planning strategies can also contribute to healthier food environments. For example, promoting access to fresh produce through farmer’s markets, community gardens, and affordable grocery stores can reduce reliance on UPFs. Subsidizing healthier food options and imposing taxes on UPFs may further shift consumer behavior toward more healthful choices [131].

## 7. Conclusions

This review elucidates the substantial and multifaceted impact of UPFs on the gut microbiome and its associated health implications. Artificial sweeteners, emulsifiers, preservatives, and food additives disrupt the delicate equilibrium of the gut microbial ecosystem, reducing microbial diversity, promoting a pro-inflammatory environment, increasing intestinal permeability, and contributing to inflammation and dysbiosis [132].

These alterations are associated with adverse metabolic, gastrointestinal, and neuropsychiatric outcomes, as well as an elevated risk of chronic diseases, including MetS, CVD, and colorectal cancer [98,107,108].

The evidence presented underscores the critical role of dietary quality in maintaining gut health. Transitioning towards a diet comprising minimally processed, nutrient-dense foods can mitigate the deleterious effects of UPFs and promote microbial resilience [118]. Furthermore, the incorporation of probiotics, prebiotics, and lifestyle modifications offers potential strategies for restoring gut homeostasis [120].

On a broader scale, public health policies must prioritize reducing UPF consumption through education, transparent labeling, and regulatory interventions. Future research should continue to elucidate the intricate mechanisms linking UPFs to gut microbiota alterations, aiming to develop evidence-based guidelines for healthier dietary practices. Collective efforts at individual, societal, and policy levels are imperative to safeguard intestinal health and overall well-being.

## Figures and Tables

**Figure 1 nutrients-17-00859-f001:**
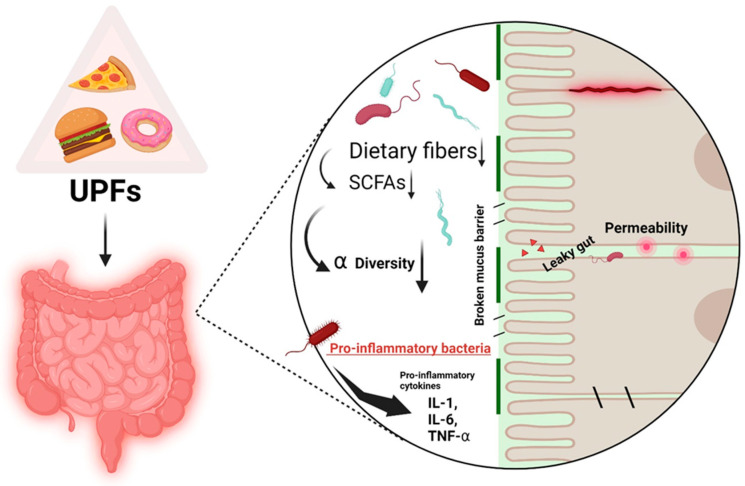
The detrimental effect of UPFs on the gut microbiome and on the gut barrier.

**Table 1 nutrients-17-00859-t001:** Strategies to mitigate the negative effects of UPFs on human health.

Strategies	Interventions
Dietary strategies	Reduction in UPF consumption Increase in dietary fiber daily intake Incorporation of fermented foods (yogurt, kefir, sauerkraut, kimchi, and miso) in dietary plans
Probiotic and prebiotic supplementation	Probiotic supplementation (*Lactobacillus rhamnosus*, *Bifidobacterium longum*, and *Saccharomyces boulardii*) Prebiotic supplementation (inulin, FOS, and GOS) Next-generation probiotics (i.e., *Akkermansia muciniphila)*
Healthy lifestyle adoption	Regular physical activity Stress management (mindfulness, meditation, and cognitive–behavioral therapy) Increase in sleep quality and consistency
Implementation of public health policies	Educational campaigns highlighting healthy dietary patterns Food labeling and marketing regulations: transparent labels, restrict marketing of UPFs targeting children Food environment interventions: provision of minimally processed foods in schools, the promotion of fresh products access, and the implementation of taxes on UPFs

Abbreviations: UPFs; ultra-processed foods; FOS, fructooligosaccharides; GOS, galactooligosaccharides; SFCA, short-chain fatty acid.

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
