# Peer review of "The Detrimental Impact of Ultra-Processed Foods on the Human Gut Microbiome and Gut Barrier"

_nutrients, 2025, doi:10.3390/nu17050859_

Round 1

Reviewer 1 Report

Comments and Suggestions for Authors

The article refers to the important  issue of the adverse impact of the ultra-processed  food  on human health. This complex problem  are presented clearly and with substantial scientific evidence. Important part of the work is to set a strategy to nitigate unfavourable effects of UPFs on health.

The review is well prepared in terms of content and editorial.

Author Response

Comment #1:

The article refers to the important issue of the adverse impact of the ultra-processed food on human health. This complex problem are presented clearly and with substantial scientific evidence. Important part of the work is to set a strategy to mitigate unfavourable effects of UPFs on health. The review is well prepared in terms of content and editorial.

Response #1:

Dear Reviewer,

we sincerely appreciate your thoughtful and positive feedback on our manuscript. We are grateful for your recognition of the importance of addressing the adverse impact of ultra-processed foods on human health and for acknowledging the clarity and scientific rigor of our presentation. Your kind words regarding the structure and editorial quality of our review are truly encouraging. Furthermore, we are pleased that you found the proposed strategies for mitigating the negative effects of ultra-processed foods to be a valuable aspect of our work. We deeply appreciate your time and effort in reviewing our manuscript and providing constructive insights. Thank you once again for your valuable comments and support.

Reviewer 2 Report

Comments and Suggestions for Authors

I have some minor comments.

Line 22, associated with

Line 93, please give more example of UPF. For example, packaged snacks, sugary drinks, fast food, and ready-to-eat meals, etc.

β-diversity is a measure of dissimilarity between samples (different samples or samples at different timepoints). Line 181. For example, beta-diversity is a measure of microbial composition between two samples.

Is section 3.2 about microbial metabolite and its function? Lines 204 to 269 should be separated into several paragraphs based on the context. A subtitle may be provided. From line 225, the content does not connect to the previous paragraph.  

Please distinguish the use of microbiome vs microbiota appropriately.

Line 222, “the” may be removed. Please use the correct term, microbial community composition or structure.

Please rephrase lines 330-335.

If ultra-processed foods (UPF) is defined, it can be consistently used in the article.

Author Response

Comment #1:

Line 22, associated with

Response #1:

Dear Reviewer,

Thank you for your comment. We have modified the sentence from "UPFs, characterised by high content of synthetic additives, emulsifiers, and low fiber content, are associated to a decrease in microbial diversity" to "UPFs, characterised by high content of synthetic additives, emulsifiers, and low fiber content, are associated with decrease in microbial diversity."

Comments #2:

Line 93, please give more example of UPF. For example, packaged snacks, sugary drinks, fast food, and ready-to-eat meals, etc.

Response #2:

We appreciate your suggestion. Additional examples of ultra-processed foods (UPFs), including packaged snacks, sugary drinks, fast food, and ready-to-eat meals, have been incorporated to enhance clarity and comprehensiveness.

Comment #3:

β-diversity is a measure of dissimilarity between samples (different samples or samples at different timepoints). Line 181. For example, beta-diversity is a measure of microbial composition between two samples.

Response #3:

Thank you for your observation. We have revised the text to explicitly define β-diversity as a measure of microbial composition dissimilarity between two samples, ensuring greater clarity for readers.

Comment #4:

Is section 3.2 about microbial metabolite and its function? Lines 204 to 269 should be separated into several paragraphs based on the context. A subtitle may be provided. From line 225, the content does not connect to the previous paragraph.

Response #4:

We appreciate your insightful comment. The section 3.2 is about functions and metabolites of the human gut microbiome. To improve readability, we have reorganized this section by separating it into multiple paragraphs based on context. Additionally, we have introduced a subtitle to enhance structural coherence. Furthermore, we have revised the transition from line 225 to ensure a logical connection with the preceding paragraph.

Comment #5:

Please distinguish the use of microbiome vs microbiota appropriately.

Response #5:

Thank you for your valuable comment. As stated in lines 39–41 of the Introduction, "The gut microbiome represents the genomic material of the gut microbiota, although the two terms will be used interchangeably here, due to the equal relevance of microbes and their genomes." Based on this clarification, we have intentionally used "microbiome" and "microbiota" interchangeably throughout the manuscript. However, we appreciate your suggestion and will ensure that the usage remains consistent and contextually appropriate.

Comment #6:

Line 222, “the” may be removed. Please use the correct term, microbial community composition or structure.

Response #6:

We appreciate your detailed feedback. The article has been updated to remove "the," and the correct term, "microbial community composition" or "structure," has been applied accordingly.

Comment #7:

Please rephrase lines 330-335.

Response #7:

Thank you for your suggestion. This section has been rephrased to improve clarity and readability while maintaining the intended meaning.

Comments #8:

If ultra-processed foods (UPF) is defined, it can be consistently used in the article.

Response #8:

We appreciate your recommendation. The term "ultra-processed foods (UPF)" has been reviewed and is now used consistently throughout the manuscript to enhance clarity and coherence. We sincerely appreciate your valuable feedback, which has significantly contributed to improving the quality of our manuscript. Thank you for your time and thoughtful suggestions.

Reviewer 3 Report

Comments and Suggestions for Authors

This article provides a thorough analysis of the impact of ultra-processed food consumption on the gut microbiome and overall health. The references are relevant, and the plagiarism index is low. The relationships and connections are clearly presented. The English language is correct and appropriately scientific.

It would be beneficial to examine whether certain populations are more sensitive to the effects of UPFs (e.g., different ethnic groups or individuals of varying ages). The study does not address how individual genetic factors, lifestyle, and geographical influences affect microbial responses to UPFs. Are there differences based on gender, age, or ethnic background?

The article also discusses regulations, but it provides few concrete examples of successful interventions from other countries. It would be valuable to include positive case studies illustrating how certain political decisions have effectively reduced UPF consumption (e.g., taxation etc.). Additionally, further discussion on which specific public health strategies have proven effective in reducing UPF consumption would strengthen the study.

Overall, this article covers an interesting and important topic.

Author Response

Comment #1:

This article provides a thorough analysis of the impact of ultra-processed food consumption on the gut microbiome and overall health. The references are relevant, and the plagiarism index is low. The relationships and connections are clearly presented. The English language is correct and appropriately scientific. It would be beneficial to examine whether certain populations are more sensitive to the effects of UPFs (e.g., different ethnic groups or individuals of varying ages). The study does not address how individual genetic factors, lifestyle, and geographical influences affect microbial responses to UPFs. Are there differences based on gender, age, or ethnic background? The article also discusses regulations, but it provides few concrete examples of successful interventions from other countries. It would be valuable to include positive case studies illustrating how certain political decisions have effectively reduced UPF consumption (e.g., taxation etc.). Additionally, further discussion on which specific public health strategies have proven effective in reducing UPF consumption would strengthen the study. Overall, this article covers an interesting and important topic.

Response #1:

Dear Reviewer,

we sincerely appreciate your thorough review and constructive feedback on our manuscript. We are pleased that you found our analysis comprehensive and the references relevant, and we value your positive comments regarding the clarity of relationships and connections presented, as well as the quality of the English language.

Regarding your suggestion to examine whether certain populations (e.g., different ethnic groups, age groups, or genders) are more sensitive to the effects of UPFs, we acknowledge the importance of this aspect. While we did not find extensive data in the literature to support distinctions in microbial responses to UPFs based on genetic factors or geographical influences, we have reported in lines 366–370 the findings of Cuevas-Sierra et al. (2021), which provide evidence of gender-based differences in the impact of UPF consumption on gut microbiota composition.

Similarly, while we recognize the relevance of including concrete examples of successful policy interventions aimed at reducing UPF consumption, our focus was primarily on outlining regulatory approaches rather than providing detailed case studies. We appreciate your suggestion and will consider incorporating selected examples where feasible to enhance the discussion.

Thank you again for your insightful comments and for recognizing the significance of our work. Your feedback has been invaluable in refining our study.